# Encouraging and Reinforcing Safe Breastfeeding Practices during the COVID-19 Pandemic

**DOI:** 10.3390/ijerph20031756

**Published:** 2023-01-18

**Authors:** Flora Ukoli, Jacinta Leavell, Amasyah Mayo, Jayla Moore, Nia Nchami, Allysceaeioun Britt

**Affiliations:** 1Department of Surgery, Meharry Medical College, Nashville, TN 37208, USA; 2Department of Dental Public Health, School of Dentistry, Meharry Medical College, Nashville, TN 37208, USA; 3Meharry Medical College, Nashville, TN 37208, USA; 4Division of Public Health Practice, School of Graduate Studies and Research, Meharry Medical College, Nashville, TN 37208, USA

**Keywords:** breastfeeding, COVID-19, African American, breastfeeding guidelines

## Abstract

Aim: Promote safe breastfeeding during the pandemic. Methods: All participants were encouraged to request safe breastfeeding education from their prenatal provider. Pregnant mothers received appropriate breastfeeding and COVID-19 safe breastfeeding education in line with the CDC’s COVID-19 breastfeeding guidelines. Data were obtained from 39 mothers attending Nashville General Hospital pediatric well-baby clinics (Group I: from December 2019 to June 2020) and 97 pregnant women attending prenatal clinics (Group II: from July 2020 to August 2021). Results: The participants’ ages ranged from 15 to 45 years, with a mean of 27.5 ± 6.2. The women in both groups were similar in age, education, employment, and breastfeeding experience. They were equally unlikely to use face masks at home even while receiving guests or holding their babies. Although 121 (89.0%) women claimed face mask use while shopping, the rate for never doing so was 7 (18.0%) vs. 8 (8.3%) (*p* < 0.006) for Groups I and II, respectively. Safe practices included limited outing (66 (48.5%)), sanitized hands (62 (45.6%)), restricted visitors (44 (32.4%)), and limited baby outing (27 (19.9%)), and 8 (8.3%) in Group II received COVID-19 vaccinations. About half described fair and accurate COVID-19 safe breastfeeding knowledge, but 22 (30.1%) of them claimed they received no information. Breastfeeding contraindication awareness for Groups I and II were as follows: cocaine = 53.8% vs. 37.1%, *p* < 0.06; HIV = 35.9% vs. 12.4%, *p* < 0.002; breast cancer = 17.9% vs. 16.5%; and COVID-19 with symptoms = 28.2% vs. 5.2%, *p* < 0.001. The information source was similar, with family, friends, and media accounting for 77 (56.6%) of women while doctors, nurses, and the CLC was the source for 21 (15.4%) women. Exclusive breastfeeding one month postpartum for Groups I and II was 41.9% and 12.8% (*p* < 0.006), respectively. Conclusion: The mothers were not more knowledgeable regarding breastfeeding safely one year into the COVID-19 pandemic. Conflicting lay information can create healthy behavior ambivalence, which can be prevented by health professionals confidently advising mothers to wear face masks when breastfeeding, restricting visitors and outings, and accepting COVID-19 vaccination. This pandemic remains an open opportunity to promote and encourage breastfeeding to every mother as the default newborn feeding method.

## 1. Introduction

The COVID-19 virus outbreak first reported in Wuhan, China rapidly developed into a pandemic, with limited knowledge of its biology and transmission. The first COVID-19 case was reported on 2 January 2020, but the World Health Organization first reported pneumonia of an unknown origin on 31 December 2019. Pandemic control measures that included high standards of personal hygiene, proper handwashing with soap or sanitizer, and very strict public health measures such as lockdowns, flight restrictions, and social distancing were disseminated around the globe [1,2,3]. The Centers for Disease Control and Prevention COVID-19 guidelines implemented by the National and State Department of Health echoed these same measures, which included lockdowns and stay home directives for all non-essential workers [4]. The very first control measure worldwide was the stay home lockdown. Special guidelines were developed to protect healthy pregnant and breastfeeding mothers, as well as women who were ill or contracted the virus [5]. Worldwide and national agencies immediately recommended the usual public health guidelines to control the spread of respiratory infectious disease, such as the use of face masks [6], mass immunization [7,8], and regular testing [9]. Strategies for successful health prevention and promotion programs may continue to evolve but cannot shift from the basic ecological model that involves actions at the individual, interpersonal, organizational, community, and policy levels [10]. While critics have accused proponents of lifestyle interventions of promoting a victim-blaming ideology by neglecting the importance of social influences on health and disease, the need to focus attention on both individual and societal factors cannot be overemphasized [11].

COVID-19 disparities exist, such as how African Americans, who are only 1.1 times at risk of infection, are 2.4 times more likely to be hospitalized and 1.7 times more likely to die of COVID-19 compared with white individuals. This risk is exceeded by the American Indian and Hispanic populations [12]. In addition to ethnicity and race, pregnancy also posed a higher risk of severe illness when contracting viruses such as COVID-19 [5]. In utero mother-to-child transmission of this virus is unlikely, but newborns are susceptible to person-to-person spread. Since the virus has not been detected in amniotic fluid or breastmilk, and breastmilk protects newborns and infants against many illnesses, it is recommended that mothers who have COVID-19 continue breastfeeding. Therefore, mothers with COVID-19 must wear a face mask and wash their hands before breastfeeding. Sick mothers who express breastmilk to maintain a milk supply should also wash their hands before touching any pump or bottle parts or consider a person without COVID-19 feeding expressed breastmilk to the infant [5]. Mothers, particularly working mothers and minority and low-income mothers who already face challenges with meeting their breastfeeding goals, will require instructions and additional support from their health care providers to meet both breastfeeding and COVID-19 protection goals [13].

COVID-19 case rates differ across workplace settings, directing attention to social inequalities and concern for vulnerable workers. Elevated risk has been reported among males, who dominate the construction, retail, and wholesale manual workforce in South Asia [14], Latino and black workers in manual labor and in-service occupations among Californian workers [15], the least skilled occupations, such as security guards, taxi drivers, and restaurant chefs [16], and health care workers [17]. Health care workers have recorded comparable low rates, probably because of rigorous implementation and enforcement of infection control and occupational safety measures [14,17]. Social and workplace inequity similarly impact breastfeeding rates as a result of variations in policies across industries, with women in service-oriented industries such as accommodation, restaurants, and retail reporting the least breastfeeding support [18], even as women in medicine [19] and public health [20] continue to experience workplace barriers including time constraints, lack of proper lounges that provide privacy and refrigeration for storing expressed milk, supervisor support, and adequate maternity leave. One third of working women, irrespective of socioeconomic status or race, succeed in combining exclusive breastfeeding (EBF) and work, highlighting the need for flexible workplace support [13], paid maternity leave [21], and extended break times, in addition to flexible work schedules [22]. Since the benefits of breastfeeding and breastmilk include protection from viral and bacterial infections [23,24], breastfeeding promotion should emphasize the additional benefit of COVID-19 protection for infants. The immunoglobulin content and anti-inflammatory properties of maternal antibodies necessary to ward off viral infections are present in breastmilk [25]. As more mothers become less familiar with breastfeeding, it should be standard prenatal practice to provide all mothers breastfeeding classes, irrespective of their proposed infant feeding plan [22]. Mothers at risk of not breastfeeding, such as low-income and African American mothers, will also require additional postnatal lactation and doula services to achieve adequate breastfeeding goals [26]. The breastfeeding promotion role of prenatal health professionals is an important cornerstone for success [27,28,29,30,31,32].

Breastfeeding is acclaimed as the optimal infant feeding method that provides significant mother and baby benefits. By the end of the 20th century, formula was entrenched as equivalent if not superior to breastmilk, especially if returning to work [33,34]. Misconceptions and negative attitudes [34,35] are counteracted by the Baby-Friendly Hospital Initiative (BFHI)’s ‘Ten Steps to Successful Breastfeeding’ [36], which is now supported by maternal and pediatric medical associations [35,37,38,39], the U.S. Preventive Services Task Force [40], and the Centers for Disease Control and Prevention [41,42]. Breastmilk plays a vital role in the development of the neonatal immune system and protection from infection by respiratory viruses, including the SARS-CoV-2 virus [43]. Promoting breastfeeding during the pandemic is therefore justified since the virus has not been detected in breastmilk, and there has been no evidence of vertical transmission [44]. It is consistently agreed that breastfeeding should be encouraged throughout the COVID-19 pandemic, and mother and infant dyads should be cared for together, except in instances where the mother is too ill to breastfeed [45].

This COVID-19 breastfeeding guideline adherence intervention focused on the impact of the individual level of health promotion on both breastfeeding and COVID-19 safety behaviors while acknowledging the important role of organizational support. The goal of this study was to increase the number of minority mothers who breastfeed adequately and safely during the COVID-19 pandemic by including COVID-19 breastfeeding guideline information in an ongoing breastfeeding promotion program. The PRECEDE-PROCEED Planning Model provided a conceptual framework where health interventions begin with socio-ecological assessments of the health issue in the target population [46,47]. Although the focus is on the individual level of the socio-ecological model, the importance of the other four interrelated levels of influence on health behaviors, namely interpersonal, community, organizational, and public policy, are acknowledged [11,48,49]. This intervention will support pregnant women by providing unbiased breastfeeding and COVID-19 safety information and give them practical strategies for success.

## 2. Materials and Methods

The study targeted African American, low-income, and minority mothers and pregnant women who received prenatal care at the Nashville General Hospital and Mathew Walker Comprehensive Health Center in Nashville, Tennessee. The intervention was developed using Community Engagement Research principles [50] in partnership with a well constituted community advisory board (CAB), whose sessions were conducted virtually. Institutional Review Board (IRB) approval was secured before implementing this intervention, which emphasized the breastfeeding guidelines for COVID-19 cases, preexisting chronic medical conditions, HIV, other infections, mental illness, and alcohol and drug use disorders. An adequate informed consent process was administered by a lactation counselor, and each participant signed IRB-approved consent forms before enrolment.

Phase 1: Health Behavior Assessment. This phase was limited to women attending the well-baby clinic with an infant less than a year old being included in the survey. Mothers of babies older than one year were excluded from participating. Thirty-nine women with infants born after 1 January 2020 who signed IRB-approved informed consent were enrolled at Meharry Medical Practice pediatric clinic to complete a short COVID-19 Breastfeeding Guideline Assessment Survey. Standard demographics, pre-existing medical conditions, and family size information were collected by a trained interviewer. COVID-19 stay-at-home and individual COVID-19 safety behaviors were ascertained by using polar questions with the yes or no answer format. COVID-19 status at enrollment was either positive, negative, or unknown for those who had not tested. Participants indicated if they breastfed at birth, 3 months, 6 months, and 12 months, providing the breastfeeding frequency and duration, breastmilk expression frequency and duration, and the frequency and volume of infant feeds with either expressed breastmilk or formula. They completed the survey within 30–45 min by interview and received a USD 20 gift card in appreciation of their time. Responses by these mothers (Group I) at this phase indicated gaps in knowledge and behavior lapses regarding COVID-19 breastfeeding safety that were then emphasized in the information provided for pregnant women in Phase 3 of this project. The Group I mothers were either pregnant or just had a baby at the onset of the COVID-19 pandemic.

Phase 2: Intervention Development. Unsupportive hospital practices and policies translate into barriers for women at high risk of not breastfeeding [40], which can be overcome by physicians emphasizing breastmilk superiority over formula. Irreversible actions during the prenatal period, at delivery, and within days of delivery can potentially compromise exclusive breastfeeding (EBF) and potentially complicate the efforts of pediatricians who support and promote breastfeeding. In light of the urgency of the COVID-19 pandemic, a draft intervention plan was tabled before a well-constituted CAB for discussion and advice, and their input was used to develop this COVID-19 safe breastfeeding intervention Figure 1. The three components of the intervention were as follows:(i)Participants received a six-panel breastfeeding brochure with a two-panel COVID-19 safe breastfeeding insert (Appendix A).(ii)Participants were advised to discuss breastfeeding during the pandemic with their obstetrician and to request referral to a certified lactation counselor for breastfeeding evaluation and training.(iii)Participants were encouraged to call the program educator or PI for breastfeeding support as needed.

Phase 3: Program Implementation Feasibility and Evaluation. Flyers were displayed in waiting rooms and handed to pregnant women attending prenatal clinics at the Nashville General Hospital and the Mathew Walker Comprehensive Health Center in Nashville, Tennessee. Inclusion criteria included being African American, having a low income, or being a minority while at least 18 years old and at least 5 months pregnant. Women in the late second or third trimester who read and signed informed consent were enrolled in the study (Group II). They understood that they would complete a pre-intervention survey, attend follow-up visits at 1 month, 3 months, and 6 months postpartum, have the weight and length information of their babies available at the follow-up, and that they would receive a USD 20 gift card and other study gifts at each completed study visit. The voluntary nature of participation was emphasized, and the participants knew they could withdraw at any time without any penalty, did not have to plan to breastfeed, and that there was no penalty for not breastfeeding or adhering to COVID-19 safety directives. They understood that average or approximate estimates of breastfeeding frequency, duration, and feed volumes were good enough, as accurate recall would be a challenge. Participants were not required to take the COVID-19 test nor the COVID-19 vaccine to enroll in this intervention. Group II mothers delivered at least one year into the pandemic.

Questionnaires:(i)Group I: Mothers recruited from December 2019 to June 2020 (early pandemic period).

**COVID-19 Breastfeeding Guideline Assessment Survey:** Demography, medical history, and breastfeeding items from CDC Breastfeeding Survey [39] and BFHI Questionnaire for Breastfeeding Mothers [51], COVID-19 safety guideline items, COVID-19 status, and stay home history.
(ii)Group II: Pregnant Women recruited from July 2020 to August 2021 (later pandemic period).

**Pre-intervention survey:** Demography, medical history, breastfeeding items from CDC Breastfeeding Survey [39] and BFHI Questionnaire for Breastfeeding Mothers [49], breastfeeding related interaction with their physician, CLC encounter, COVID-19 guideline items, COVID-19 status, and stay home history.

**Post-intervention survey:** CLC referral and consultation, breastfeeding class attendance, COVID-19 guideline items, breastfeeding items such as breastfeeding initiation at birth, breastfeeding frequency and duration, breastmilk expression frequency and duration, and formula or breastmilk feed frequency and volume.

Data Analysis:

The primary study outcomes were COVID-19 breastfeeding guideline adherence and EBF rates at 1 month, 3 months, and 6 months. Descriptive demographic statistics were tabulated by enrollment period, comparing mothers in the early COVID-19 pandemic period (Group I) to mothers at least one year into the COVID-19 pandemic period (Group II). Comparative statistical tests between mothers in Groups I and II and across demographic subgroups were two-sided using a 5% significance level and performed using the two-group T-test for continuous variables or the Chi-square test or Fisher’s exact test for discrete data as appropriate. The SPSS software (version 26) was used to conduct all statistical analyses.

## 3. Results

The age range for 136 enrolled mothers was 15–45 years, with a mean of 27.5 ± 6.2 years, and mean ages of 39 in Group I and 97 in Group II of 29.5 ± 7.1 years versus 26.7 ± 5.7 years, respectively (*p* < 0.02). Group I mothers were more likely to be married (13 (33.3%) versus 17 (17.5%), *p* < 0.04) and unemployed (16 (41.0%) versus 24 (24.7%), *p* < 0.04). Among working mothers, 11% planned to return to work by 4 weeks postpartum, 76% planned to return in between 4 and 12 weeks, and 13% planned to return by at least 13 weeks postpartum. About half of the study population had no more than a high school education, and just over two thirds were on government nutrition assistance, namely the Supplemental Nutrition Assistance Program (SNAP) or Special Supplemental Nutrition Program for Women, Infants, and Children (WIC). (Table 1). The mothers worked in offices or schools, (23.5%), stores, warehouse, or factories, (23.5%), hospital or medical facilities, (16.3%), at home (15.3%), restaurants or fast food, (12.2%), and emissions. (9.2%). Overall, 57 (58.2%) working mothers claimed to have privacy to express breastmilk at work (10 (41.7%) versus 47 (64.4%), *p* < 0.05 for Groups I and II, respectively). Living with at least one preexisting condition such as hypertension, diabetes, or asthma was reported by 30 (22.1%) mothers, and 14 (10.3%) admitted marijuana use.

Regarding reported breastfeeding experience, 49 (36.0%) witnessed their mother, sister, or aunt breastfeeding, and 48(35.3%) witnessed a friend, while 43 (31.6%) never saw anyone breastfeeding. Of the 104 mothers with a previous baby, 33 (31.7%) breastfed for less than a month, 30 (28.8%) for 1–5 months, and 21 (20.2%) for 6 months or longer. Meanwhile, 69 (50.7%) had not decided on a breastfeeding plan, 30 (22.1%) planned to breastfeed for 3–6 months, and 37 (27.2%) planned to do so for 6 months or longer. Formula use at birth and any time before 6 months was the plan for 106 (78.0%) mothers, 11 (8.1%) planned to introduce formula at 6 months, and 19 (14.0%) planned to not use any formula. Referral to the WIC program was reported by 48 (65.8%) mothers, 34 (46.6%) were referred to a CLC, and 22 (30.1%) received information about breastfeeding safely during the COVID-19 pandemic. In addition, 78 (57.4%) described fair and accurate COVID-19 breastfeeding safety tips, 21 (15.4%) cited health providers as their information source, and 22 (30.1%) claimed they did not receive COVID-19 safe breastfeeding information (Table 2).

Overall, 121 (89.0%) mothers reported face mask use while shopping, and the rate for never doing so for Groups I and II was 7 (18.0%) versus 8 (8.3) (*p* < 0.006), respectively. Face mask use at home was reported to be unlikely by 114 (83.8%) mothers, but they would do so while receiving guests (14 (10.3%)) and while holding their babies (4 (2.9%)). COVID-19 safety practices included limited mother outing (66 (48.5%)), sanitizing hands regularly (62 (45.6%)), restricted visitors to the house (44 (32.4%)), and limited baby outings (27 (19.9%)), and 8 (8.2%) Group II mothers received COVID-19 vaccinations. COVID-19 testing was not performed by 71 (52.2%) mothers, while 56 (41.2%) tested negative and 8 (5.8%) were positive (Table 3).

Awareness of health and drug breastfeeding contraindications among the mothers in Groups I and II were 24 (61.5%) vs. 33 (34.0%) (*p* < 0.001) and 28 (71.8%) vs. 53 (54.6%) (*p* < 0.01), respectively. Knowledge for the mothers about specific contraindications were as follows for cocaine (53.8% vs. 37.1%, *p* < 0.06), HIV (35.9% vs. 12.4%, *p* < 0.002), breast cancer (17.9% vs. 16.5%), and COVID-19 with symptoms (28.2% vs. 5.2%, *p* < 0.001). The Group I and II mothers who had some or the correct idea about the definition of exclusive breastfeeding totaled 29 (74.4%) versus 35 (36.1%) (*p* < 0.001), respectively (Table 4).

Exclusive breastfeeding one month postpartum for Groups I and II was reported by 41.9% and 12.8% (*p* < 0.006) of mothers, while the results were 33.3% and 4.5% (*p* < 0.007) for 3 months and 20.0% and 0% for 6 months, respectively. Providing the baby breastmilk one month postpartum for Groups I and II was reported by 67.7% and 56.4% (*p* < 0.03) of mothers, while the results were 55.6% and 31.8% (*p* < 0.02) for 3 months and 46.7% and 33.3% for 6 months (Table 5).

## 4. Discussion

Although COVID-19 is an acute infectious disease, and cancer is primarily a chronic condition, both benefit from multilevel public health strategies directed at the patient, provider, community, and healthcare policy, thus sharing a common set of barriers to care impacted by social determinants of health. Lessons learned from decades of cancer prevention and control, such as the impact of rapid integration of research evidence into practice, is a proven strategy that can similarly improve COVID-19 response [52]. In the same way that racial and ethnic disparities in cancer survival are largely attributable to poverty, delayed screening, differences in provider care recommendations, and lack of access to the latest treatments [53], emerging evidence indicates the same for COVID-19 deaths due to preventable underlying causes [54]. As effective strategies to treat and manage COVID-19 accumulate, the potential for disparities will increase, as access to these healthcare advances are unlikely to be equally distributed [53]. This study indicates similarly low COVID-19 breastfeeding safety knowledge and practices among mothers enrolled at the onset of the COVID-19 pandemic and those enrolled one year into the pandemic. One reason for this may be related to the absence of breastfeeding training for pregnant mothers. Most mothers in this study reported that they did not receive breastfeeding training, and only very few were referred to a CLC. It is not unlikely that mothers who declared a desire to formula feed were not referred for such counseling. Most women in this study population did not witness their mothers breastfeed, and only 36% witnessed any family member doing so. The lack of breastfeeding role models within the family might contribute to low breastfeeding rates in populations at increased risk of not breastfeeding. To overcome such a societal bias, the WHO and the CDC have since recommended that all mothers receive breastfeeding education (Step 3 of the ‘Ten Steps to Successful Breastfeeding’) instead of providing such education only to the few women who request it. If mothers were not receiving breastfeeding training routinely, it was unlikely that they would have received COVID-19 safe breastfeeding directives. In this study, 70% did not receive any COVID-19 safe breastfeeding directives, and most of those who reported knowledge in this area received that information from the media, friends, and family members. Only 15% received such directives from a doctor, nurse, or CLC. There was no difference in the pattern of information sources between mothers seen at the onset and one year into the pandemic. More worrisome is the overall drop in general breastfeeding knowledge and practice among Group II mothers compared with Group I, suggesting a possible oversight in maintaining routine breastfeeding protection, promotion, and support care during the pandemic.

Mothers and their families need to be well informed about breastfeeding during COVID-19 [43,44,55]. Breastfeeding should be encouraged for all mothers, and even mothers who are too ill to breastfeed should be encouraged to express their milk [45]. They will, however, need to be supported to maintain appropriate respiratory hygiene [56] and adequate infection control measures [57]. COVID-19 safe breastfeeding knowledge and behavior in this population was similarly low across education, economic, and marital sociodemographic groups. That multiparous mothers also recorded low knowledge and safe breastfeeding practices compared with the mothers having their first babies is an indication that they did not receive necessary breastfeeding promotion support in their previous pregnancies. Adequate breastmilk initiation and flow sustenance stand firm on the concept of constant mammary gland stimulation resulting from ‘rooming in’ and ‘feeding on demand’. Mothers who were probably unaware of such concepts did not know to capitalize on the COVID-19 lockdown opportunity and maximize the opportunity to achieve their full lactation potential. Just over half of these mothers were at home, either because of the COVID-19 directive or because they were initially unemployed. Only one third of those who had planned to breastfeed claimed they were able to breastfeed more by being at home, while staying at home did not make any difference to those who had not planned to breastfeed. They did not seize the opportunity to breastfeed, probably because they were not aware that staying home was an advantage for their babies. This finding reflects the results from other studies, where minority and low-education mothers struggled to receive breastfeeding support while others were able to exploit the breastfeeding opportunity of the lockdown [58]. Safe breastfeeding action was also very low both in the early and later pandemic periods, with up to 80% taking their babies to public places. Most mothers did use a face mask while shopping but did not protect their babies by restricting visitors to the house nor wear a face mask while holding their babies. It would appear that they were not aware of the COVID-19 safe breastfeeding recommendations to protect their babies [59].

A Phase III randomized, multicenter, endpoint-driven, double-blind, placebo-controlled clinical trial conducted among healthy adults clearly demonstrated the efficacy and safety of the adsorbed inactivated COVID-19 vaccine [60]. The professional board of obstetrics and gynecology published a position statement to offer vaccination to pregnant women [61] and requested that medical professionals not allow breastfeeding mothers to face the decision to breastfeed or vaccinate alone [62]. Research has also demonstrated the effectiveness and safety of COVID-19 vaccination among breastfeeding mothers, with only minimal disruption of lactation or adverse effects on the breastfed child [63], providing anti-SARS-CoV-2-specific IgA and IgM antibodies that pass into the breastmilk and protect the breastfed baby [64,65]. It is important for anyone handling a baby, including mothers, to be vaccinated and masked [65]. Although all participants in this study were informed about the importance of COVID-19 testing and vaccine safety in pregnant and lactating mothers, and they received individual encouragement by the program educator to get tested and vaccinated, only 8% of the mothers in the later pandemic period received vaccinations when they became available, and just about half of the study participants were tested for COVID-19. Vaccine uptake in this population was much lower than the 44.4% uptake for pregnant women reported in another study. They observed a lower uptake compared with those for non-pregnant women (76%) and breastfeeding mothers (55%), which was attributed to the additional stress of being pregnant and the misplaced fear of the effect of the vaccine on the fetus [66]. Inadequate professional breastfeeding support in populations at a high risk of not breastfeeding is expected. Many of the women in the study’s target population were working mothers on low wages, including essential workers, and they reported a lack of workplace support such as breastfeeding lounges or extended break periods. They denied receiving information about how to breastfeed safely during the pandemic and did not notice provisions or directives to get vaccinated while pregnant or lactating at their doctor’s offices or their workplaces.

There is a need for adequate coordination across the mother and baby care continuum. The team of physicians, nurses, and midwives should engage mothers in breastfeeding conversations early in pregnancy, actively encourage and refer them to CLCs for breastfeeding evaluation and training, and not leave mothers to make decisions about infant feeding based on the misconception of breastmilk and formula equivalence [67]. Pediatricians will readily prescribe appropriate formula for rare but real instances as required [68]. Physicians who provide prenatal care are, however, in the best position to recognize potential breastfeeding challenges, provide personalized guidance for individual circumstances [69,70], and address emerging health threats by emphasizing COVID-19-specific breastfeeding guidelines at this current time [5]. CLCs are trained to provide in-depth breastfeeding education and training and have the time to meet the added breastfeeding needs of the over 50% of mothers in the United States whose babies are inadvertently offered formula and pacifiers at birth [71]. Allowing babies to latch onto the bottle at birth is a known reason for babies experiencing challenges or distress latching to their mother’s nipple, a condition sometimes referred to as ‘nipple confusion’ [72,73]. Delivery room nurses and physicians are in the position to ensure breastfeeding initiation at birth (Step 4: ‘Ten Steps to Successful Breastfeeding’), as the skin-to-skin action by itself is insufficient breastfeeding care for the newborn. While obtaining an International Board-Certificated Lactation Consultant (IBCLC) designation is not required to provide basic delivery room breastfeeding care to mothers, prenatal and neonatal care providers do need the skills to effectively advocate for patients, especially for uneducated [74] and adolescent mothers [75]. They also need to monitor and ensure that their patients receive adequate breastfeeding support both at birth and postpartum [76,77,78]. In this study, the mothers did not report postpartum breastfeeding follow-ups. Physicians may be less inclined to conduct postpartum breastfeeding follow-ups since a majority of mothers elect to bottle feed with formula.

## 5. Conclusions

The mothers in this study recorded low breastfeeding rates with little adherence to COVID-19 safe breastfeeding practices one year into the pandemic. The pandemic remains an open opportunity to intensify breastfeeding programs, encourage every mother to breastfeed, and for hospitals and maternities to adopt breastfeeding as the default newborn feeding method. Physicians, nurses, and midwives should refer all pregnant women to CLCs for breastfeeding evaluation and training and invite the IBCLC when medically indicated. Conflicting lay information that creates ambivalence can be prevented by urgently implementing policy revisions within existing breastfeeding guidelines to address emergent threats such as the COVID-19 pandemic. Health professionals should confidently advice mothers to frequently sanitize their hands, wear face masks when breastfeeding, limit outings, restrict visitors, and take COVID-19 tests and vaccines.

## Figures and Tables

**Figure 1 ijerph-20-01756-f001:**
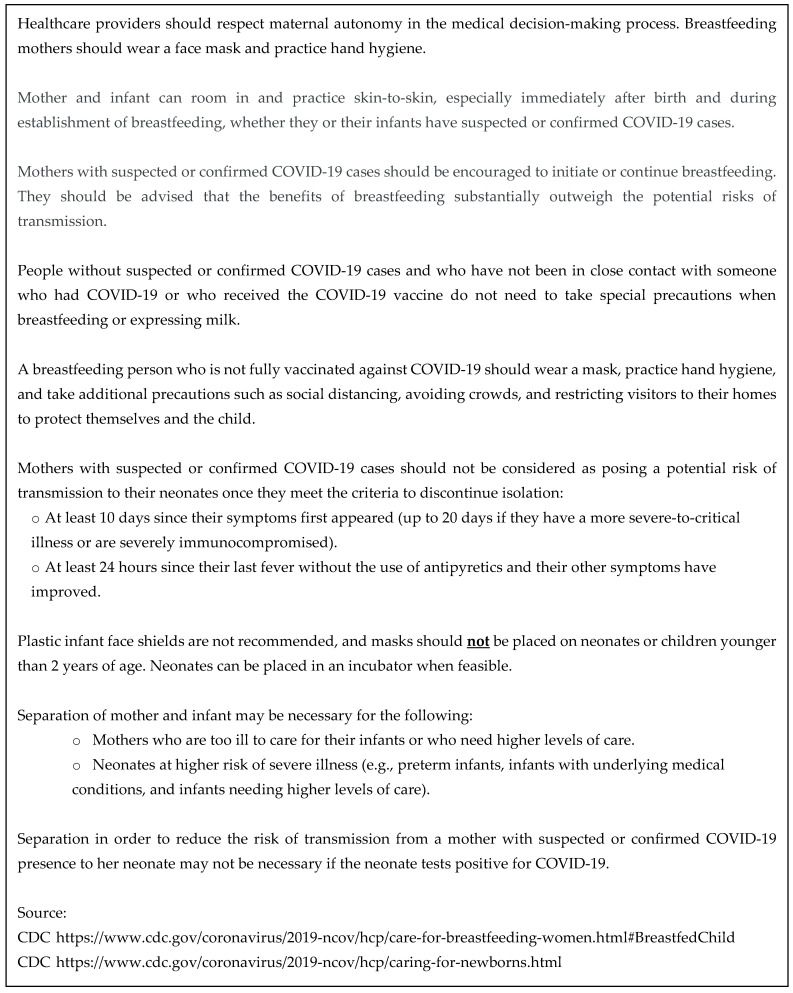
Summary of CDC and WHO SARS-CoV-2 (COVID-19) breastfeeding recommendations.

**Table 1 ijerph-20-01756-t001:** Demographic characteristics of participating mothers during the early and later COVID-19 pandemic periods.

Characteristics	Study Population		Total	
	Group I ^§^ Early Pandemic	Group II ^§§^ Later Pandemic	No.	%
Age in years *				
<21	4 (10.3)	14 (14.4)	18	13.2
21–34	25 (64.1)	74 (76.3)	99	72.8
>34	10 (25.6)	9 (9.3)	19	14.0
Education				
High School	21 (53.9)	52 (53.6)	73	53.7
Some College	5 (12.8)	22 (22.7)	27	19.8
College Degree	13 (33.3)	23 (23.7)	36	26.5
Marital Status *				
Single	26 (66.7)	80 (82.5)	106	77.9
Married	13 (33.3)	17 (17.5)	30	22.1
Employment				
Unemployed	14 (35.9)	24 (24.7)	38	27.9
Part-Time	5 (12.8)	29 (29.9)	34	25.0
Full-Time	20 (51.3)	44 (45.4)	64	47.1
Income **				
<USD 25,000	16 (41.0)	33 (34.0)	49	36.0
USD 25,000–49,999	14 (35.9)	19 (19.6)	33	24.3
>USD 50,000	8 (20.5)	13 (13.4)	21	15.4
Not Stated	1 (2.6)	32 (33.0)	33	24.3
Nutrition assistance *				
WIC ^#^ and SNAP ^##^	19 (48.7)	26 (26.8)	45	33.1
WIC or SNAP	12 (30.8)	37 (38.1)	49	36.0
None	8 (20.5)	24 (35.1)	42	30.5
Work Return Plan				
<4 Weeks	4 (10.3)	7 (7.2)	11	8.1
4–7 Weeks	8 (20.5)	27 (27.8)	35	25.7
8–12 Weeks	9 (23.1)	32 (33.0)	41	30.1
≥13 Weeks	5 (12.8)	8 (8.2)	13	9.6
Not Working	13 (33.3)	23 (23.7)	36	26.5
Daycare Use Plan ***				
Yes	4 (10.3)	29 (29.9)	33	24.3
No	35 (89.7)	44 (45.4)	79	58.1
Undecided	0 (0.00)	24 (24.7)	24	17.6

^§^ Group I: Meharry Well-Baby Clinic, early pandemic period (from December 2019 to June 2020). ^§§^ Group II: Meharry Prenatal Clinic, later pandemic period (from July 2020 to August 2021). ^#^ WIC: Special Supplemental Nutrition Program for Women, Infants, and Children. ^##^ SNAP: Supplemental Nutrition Assistance Program. * *p* < 0.05. ** *p* < 0.01. *** *p* < 0.001.

**Table 2 ijerph-20-01756-t002:** Preparation for safe breastfeeding among participating mothers during the early and later COVID-19 pandemic periods.

Characteristics	Study Population		Total	
	Group I Early Pandemic	Group II Later Pandemic	No.	%
Privacy at work to breastfeed				
Among working mothers *	10 (41.7)	47 (64.4)	57	58.2
Breastfeeding experience				
Was breastfed	13 (33.3)	28 (28.9)	41	30.1
Witnessed mother	9 (23.1)	17 (17.5)	26	19.1
Witnessed mother or relatives	12 (30.8)	37 (38.1)	49	36.0
Witnessed friend	16 (41.0)	32 (33.0)	48	35.3
Witnessed anybody	24 (61.5)	69 (71.1)	93	68.4
Breastfed previous baby				
<1 month	9 (23.1)	24 (24.7)	33	24.3
1–5 months	9 (23.1)	21 (21.6)	30	22.1
≥6 months	9 (23.1)	12 (12.4)	21	15.4
No previous baby	12 (30.8)	40 (41.2)	52	38.2
Referral				
WIC	26 (74.3)	22 (57.9)	48	65.8
CLC	19 (54.3)	15 (39.5)	34	46.6
COVID-19 breastfeeding	12 (34.3)	10 (26.3)	22	30.1
Breastfeeding goal				
None or undecided	19 (48.7)	50 (51.5)	69	50.7
3–5 months	9 (23.1)	21 (21.7)	30	22.1
≥6 months	11 (28.2)	26 (26.8)	37	27.2
Formula plan				
At birth or undecided	19 (48.7)	65 (67.0)	84	61.8
Before 6 months	10 (25.7)	12 (12.3)	22	16.2
≥6 months	5 (12.8)	6 (6.2)	11	8.1
No formula	5 (12.8)	14 (14.4)	19	14.0
^#^ COVID-19 BF knowledge.				
Fair or correct	19 (48.7)	59 (60.8)	78	57.4
Incorrect or do not know	20 (51.3)	38 (39.2)	58	42.6
Information source **				
Media, self, friend, or family	20 (51.3	57 (58.8)	77	56.6
Doctor, nurse, or CLC	4 (10.3)	17 (17.5)	21	15.4
Not recorded	15 (38.5)	23 (23.7)	38	27.9

^#^ COVID-19 breastfeeding safety guideline knowledge. * *p* < 0.05. ** *p* < 0.001.

**Table 3 ijerph-20-01756-t003:** COVID-19 breastfeeding safety guideline adherence pattern among mothers during the early and later COVID-19 pandemic periods.

Characteristics	Study Population		Total	
	Group I Early Pandemic	Group II Later Pandemic	No.	%
Facemask at home				
Never	29 (74.4)	85 (87.6)	114	83.8
Guests visiting	6 (15.4)	8 (8.2)	14	10.3
Holding baby	3 (7.7)	1 (1.0)	4	2.9
Almost always	1 (2.6)	3 (3.1)	4	2.9
Face mask shopping **				
Never or sometimes	7 (18.0)	8 (8.3)	15	11.0
Most always	32 (82.1)	89 (91.7)	121	89.0
COVID-19 safety action				
Mother outings limited	17 (43.6)	49 (50.5)	66	48.5
Hands washed or sanitized	17 (43.6)	45 (46.4)	62	45.6
Visitors restricted	11 (28.2)	33 (34.0)	44	32.4
Baby outings avoided	6 (15.4)	21 (21.6)	27	19.9
Vaccinated	0 (0.0)	8 (8.2)	8	5.9
COVID-19 testing				
Not testing	14 (35.9)	57 (58.8)	71	52.2
Negative test	21 (53.8)	36 (37.1)	57	41.2
Positive with no symptoms	0 (0.0)	1 (1.0)	1	0.7
Positive with symptoms	4 (13.0)	3 (3.1)	7	5.1

** *p* < 0.01.

**Table 4 ijerph-20-01756-t004:** Breastfeeding contraindication awareness among participating mothers.

Breastfeeding Knowledge Topic	Study Population		Total	
	Group I 2019–2020	Group II 2020–2021	No.	%
Health contraindications ***				
Yes	24 (61.5)	33 (34.0)	57	41.5
No	15 (38.5)	28 (28.9)	43	31.6
Do not know	0 (0.0)	36 (37.1)	36	26.5
Drug contraindications **				
Yes	28 (71.8)	53 (54.6)	81	59.6
No	11 (28.2)	26 (26.8)	37	27.2
Do not know	0 (0.0)	18 (18.6)	18	13.2
Specific health contraindication				
HIV **	14 (35.9)	12 (12.4)	26	19.1
Breast cancer	7 (17.9)	16 (16.5)	23	16.9
COVID-19 with symptoms ***	11 (28.2)	5 (5.2)	16	11.8
COVID-19 with no symptoms **	6 (15.4)	3 (3.1)	9	6.6
Mental illness (severe) **	6 (15.4)	2 (2.1)	8	5.9
Specific drug contraindication				
Cocaine *	21 (53.8)	36 (37.1)	57	41.9
Heroin	17 (43.6)	33 (34.0)	50	36.8
Marijuana	11 (28.2)	23 (23.7)	34	25.0
Opioid or oxycodone *	13 (33.3)	16 (16.5)	29	21.3
Chemotherapy ***	9 (23.1)	3 (3.1)	12	8.8
Exclusive Breastfeeding ***				
Some or correct idea	29 (74.4)	35 (36.1)	64	47.0
Wrong idea or do not know	10 (25.6)	62 (63.9)	72	52.9

* *p* < 0.05. ** *p* < 0.01. *** *p* < 0.001.

**Table 5 ijerph-20-01756-t005:** Comparing breastfeeding rates among mothers in the early (2019–2020) and later (2020–2021) COVID-19 pandemic periods.

Breastfeeding Patterns	Group I Early Pandemic	Group II Later Pandemic	Totals ^§^	
			No.	%
1 month postpartum	n = 31	n = 39	n = 70	
Breastmilk provided *	21 (67.7)	22 (56.4)	43	61.4
Exclusive breastfeeding **	13 (41.9)	5 (12.8)	18	25.7
Nursing at the breast *	19 (61.3)	15 (38.5)	34	48.6
3 months postpartum	n = 27	n = 22	n = 49	
Breastmilk provided *	15 (55.6)	7 (31.8)	22	44.9
Exclusive breastfeeding **	9 (33.3)	1 (4.5)	10	20.4
Nursing at the breast	13 (48.1)	5 (22.7)	18	36.7
6 Months postpartum	n = 15	n = 12	n = 27	
Breastmilk provided	7 (46.7)	4 (33.3)	11	40.7
Exclusive breastfeeding	3 (20.0)	0 (0.0)	3	11.1
Nursing at the breast	6 (40.0)	4 (33.3)	10	37.0

* *p* < 0.05. ** *p* < 0.01. ^§^ Totals based on follow-up completed by December 2021.

## Data Availability

The data presented in this study are available on request from the corresponding author. The data are not publicly available at this time as this is an ongoing project.

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
