# Peer review of "Encouraging and Reinforcing Safe Breastfeeding Practices during the COVID-19 Pandemic"

_ijerph, 2023, doi:10.3390/ijerph20031756_

Round 1
Reviewer 1 Report
The topic of the manuscript entitled „Encouraging and Reinforcing Safe Breastfeeding Practices During COVID-19 Pandemic” covers a topic of research that has received increased attention in recent years, namely the importance of breastfeeding promotion as a facilitator for adequate breastfeeding practices. The particularities addressed by the research in relation to the application of COVID-19 safe breastfeeding practices by lactating women contribute to its originality. The results of the work are valuable and can be capitalized on to continue developing customized programs that promote and support breastfeeding in various epidemiological situations, with special emphasis on the needs of African American mothers.
The manuscript is generally well-written. However, there are some aspects that should be addressed before a decision on publication. The manuscript needs revision focused on the specific remarks stated below.
MAJOR REMARKS:
1. 1. The authors must thoroughly revise the manner in which they present the results throughout the manuscript („Abstract section” and „Results section”). At this point, many results are difficult to be comprehended. For example, the results presented in line 22 „Although 121(89.0) claimed facemask use while shopping…”. What does number 121 describe? The number of mothers? What about the number 89.0? What does it stand for and / or what unit of measure does it have?
2. 2. The “Discussion section” requires improvements. More previous studies need to be cited by the authors to discuss the results of their study. As an example, for the result presented in lines 285-286 „Most women in this study population did not witness their mother breastfeed, and only 36% witnessed any family member doing so.”, the authors provide the following interpretation „It is therefore not surprising that many of them opt to feed their babies formula.”. This is a personal interpretation that should be replaced or at least supported by previous studies that show similar results.
3. 3. The references in the manuscript should be presented as requested by the IJERPH instructions for authors, meaning that reference numbers should be placed in square brackets [ ], and placed before the punctuation.
4. 4. It would be interesting for the authors to provide as supplementary material, images of the flyers that have been used within the intervention to promote breastfeeding during COVID-19.
MINOR REMARKS
1. Line 16: Please revise the phrase „Individualized counseling for participants was by a trained educator.”
2. Line 77: Please revise the word „defer”.
3. Lines 140-141: „by a trained research assistant”.
5. Line 221: Please delete the word „the” before the number 39.
6. Line 222: Please revise „p<02”.
7. Line 231: „claimed”.
Author Response
Thank you for all your valid comments that are highly appreciated. I have addressed them as shown below.
MAJOR REMARKS:
1). The number 121 is the frequency, that is number of mothers who claimed to use facemask. 121(89.0)- The number 89.0 is the percentage. 121/136 = 88.97%. The unit therefore is %. The result can therefore be revised as 121(89.0%).
% has been indicated as the unit in the bracket ( %) throughout the result section.
2). The statement "It is therefore not surprising that many of them opt to feed their babies formula.”. has been replaced with:
“The lack of breastfeeding role models within the family might contribute to low breastfeeding rates in populations at increased risk for not breastfeeding.”
3). References have been properly placed in square brackets [ ] before the punctuation.
4). Images of flyers and brochures will be uploaded as supplementary materials.
Will upload a flyer and the brochure insert.
MINOR REMARKS
- Line 16: Please revise the phrase „Individualized counseling for participants was by a trained educator.”
“All participants were encouraged to request safe breastfeeding education from their prenatal provider. Pregnant mothers received appropriate breastfeeding and COVID-19 safe breastfeeding education in line with the CDC COVID-19 breastfeeding guidelines.”
- Line 77: Please revise the word „defer”. “defer” spelling typo should be ‘differ’. I have revised the word to 'differ'. Will you prefer i use the word “vary”?
- Lines 140-141: „by a trained research assistant”. Has been revised to read …”by a lactation counselor”…
- Line 221: Please delete the word „the” before the number 39.
…’the’ has been deleted.
- Line 222: Please revise „p<02”.
Revised to p<0.02.
- Line 231: „claimed”.
Revised accordingly.
Reviewer 2 Report
The article deals with the very serious topic of breastfeeding with COVID-19. In most countries, protocols are different, due to lack of resources, mothers are separated from their children.
The paper is excellently done methodologically, with well-presented results and a comparative order in the discussion. References adequately accompany the text. I believe that the work is interesting to a wider audience of public health and to clinicians
Accept
Author Response
Thank you very much for your kind comments. I have made some minor revisions to address comments of other reviewers.
Reviewer 3 Report
Article Title ‘’ Encouraging and Reinforcing Safe Breastfeeding Practices During COVID-19 Pandemic’’.
I’m a little confused when reading it, I’m not sure what is meant by ‘’The goal of this study was to increase the number of minority mothers who breastfeed adequately and safely during the pandemic, by including COVID-19 breastfeeding guideline information in an ongoing breastfeeding promotion program’’. Need some clear statements here. The main contribution is not made so clear, perhaps through more discussion of literature and identification of gaps in our understanding would help to clarify this.
Materials and Methods: I think it would be useful to have some research questions / hypotheses in this section in order to allow clearer interpretation of the results. The over-arching research aim is clear in the introduction, but I believe it is essential to also have some guiding research questions / hypotheses. Given this would be post-hoc, it would not be appropriate to retrofit research questions. However, I suspect the researchers had questions/hunches before data collection and these could be displayed here – this will require some thought from the researchers. This research is a health education intervention, but the findings include incomparable situations such as group I and group II. In this study, it would be better to take the measurements of group II for behavioral change after health education according to the follow-up months (Line 180 181- They understood that they would complete a pre-intervention survey, attend follow-up visits at 1-month, 3-month, and 6-month postpartum, 181 have the weight and length information of their baby available at follow-up…..). I took quite some time deliberating on this manuscript
Ultimately, I think the topic of this paper is important, but I think in its current form the contribution this paper can make to the literature is limited. The authors should reconsider the presentation of the results.
Author Response
Thank you for your kind comments. I can understand the confusion. This manuscript is going the be the first to report the findings of our ongoing project. We are still recruiting and following up mothers. Stay tuned for more results. I wish the following will provide enough explanations to your comments.
Response:
The project long term goal is to increase safe breastfeeding during the pandemic. We were conducting standard breastfeeding promotion before COVID-19, and we expanded our education to include the COVID-19 Safe Breastfeeding Guidelines from CDC. We wanted mothers to advocate for themselves and receive as much safe breastfeeding education and support as possible. This manuscript has only addressed one aspect of the entire issue at hand.
We attempted to observe changes in breastfeeding and safe breastfeeding behavior among minority/African American/low-income women at the onset of the pandemic and one year later.
The questions on our minds were as follow:
1: Are minority mothers getting appropriate breastfeeding education reflected in their breastfeeding behavior?
2: Are these mothers receiving education about how to breastfeed safely during the COVID-19 epidemic?
Mothers in group I already had their baby at the onset of the pandemic. All they would have received will be standard breastfeeding education from their prenatal doctor, if any. They completed the breastfeeding survey that including breastfeeding action at birth, 3-months, 6-months and 1-year.
Mothers in group II were pregnant during the pandemic and this project encouraged them to breastfeed safely and to ask their prenatal doctor to educate them about safe breastfeeding during the pandemic. We were hoping they will receive more safe breastfeeding education than the mothers in group I. They completed the same breastfeeding survey and we contacted them during the follow-up at birth, 3-months, 6-months, and 1-year.
This manuscript is only reporting observed differences in breastfeeding rates and observed differences in COVID-19 safe breastfeeding actions between these two sets of mothers. We wanted to see if there is a difference among the mothers at the onset of the pandemic and the mothers one year later.
We plan to publish the detailed follow-up information of group II mothers in a subsequent report. We are not testing our intervention. We are trying to see if prenatal breastfeeding education had incorporated COVID-19 safe breastfeeding education one year into the pandemic.
Reviewer 4 Report
Good morning,
I congratulate the authors for the initiative to investigate this topic, which is also one of those I am working on with a research group, so it is of great interest to me.
Having said that, I would like to express my opinion regarding the methodology section, which could be improved by:
- specifying the criteria for inclusion and exclusion of the sample.
- specifying the methodology for the analysis of the qualitative data and the thematic axes that articulate it
- specify the criteria used to determine the saturation of (qualitative) information
I look forward to receiving answers to these questions in order to move forward in the revision process.
Kind regards
Author Response
Thank you very much for your thorough review. You will be relieved to know we did not collect qualitative data. Study participants completed a survey as described in the methods section.
We partnered with a CAB to develop the content and design of the education intervention.
Criteria for inclusion and exclusion has been added to the Method section.
The study was limited to African American, low-income and minority women who attend the Nashville General Hospital and the affiliated Mathew Walker Health Center.
Group I: This phase was limited to women attending the well-baby clinic with an infant less than a year old to be included in the survey. Mothers of babies older than one year were excluded from participating.
(Mothers in the pediatric well-baby clinic who delivered a baby between December 2019 to June 2020.)
Group II: Inclusion criteria included being Africa American, low-income, or minority, at least 18 years old, and at least 5 months pregnant.
(Pregnant Mothers attending the prenatal clinic from July 2020 – August 2021.)
Exclusion Criteria: None minority women who either did not have a baby between December 2019 to June 202, or were not pregnant between July 2020 to August 2021.
Methodology for the analysis of the qualitative data and the thematic axes that articulate it:
Response: We did not collect qualitative data.
Specify the criteria used to determine the saturation of (qualitative) information
Response: We did not collect qualitative data.
Round 2
Reviewer 3 Report
Dear authors,
Thank you for your kind responses. I can better understand the content of the article now. However, some readers may think like me when reading for the first time. For this reason, it would be great if you state in the main text that the research is the first findings of an ongoing project.